# A Systematic Literature Review on Water Insecurity from an Oregon Public Health Perspective

**DOI:** 10.3390/ijerph17031122

**Published:** 2020-02-10

**Authors:** Cordelia Schimpf, Curtis Cude

**Affiliations:** Public Health Division, Oregon Health Authority, Portland, OR 97232, USA; cordelia.schimpf@gmail.com

**Keywords:** household water insecurity, human right to water, public health, social determinants of health, water access, water insecurity, water poverty index, water scarcity, water security

## Abstract

This paper systematically reviews existing United States-based water insecurity literature with the goal of understanding the evidence base for developing public health water insecurity intervention strategies in Oregon. The authors conducted the systematic literature review using an adjusted PRISMA reporting checklist to document the review process. Results find 11 public health-related water insecurity interventions including surveillance practices and indicator and policy development. Research on water insecurity health impacts and solutions is still an emerging field. Nevertheless, state agencies perceive a risk to communities from inadequate safe water and are taking steps to assess and reduce these risks. From the review, strategies include improving water affordability, carrying out community education events, documenting drought risk and water loss, and tracking improvements in safe drinking water compliance. The review finds opportunities to take varied approaches that are community-specific, partnership-based and culturally relevant. Recommendations for Oregon include characterizing communities experiencing water insecurity, assessing community needs, tracking regional water scarcity and recognizing the human right to water in Oregon.

## 1. Introduction

Despite the common belief that Oregon is water-rich, population- and climate-driven pressures on water insecurity are a real concern for all people in Oregon. In this paper, we define “water insecurity” as inadequate or inequitable access to clean, safe and affordable water for drinking, cooking and sanitation and hygiene. As an ideal and opposite state, “water security” describes the conditions where water quality, quantity and access are enough to protect public health.

Three key factors affecting water security include:Climate changes that are increasing the frequency and severity of droughts, floods, wildfires and other natural disasters. These strain our aging infrastructure and expand water insecurity threats to vulnerable communities [1].Social determinants that affect access to clean and safe water such as socioeconomic conditions (for example, concentrated poverty), population distribution and community engagement.Physical determinants of water security such as drinking water and wastewater storage, treatment and delivery systems, housing status and geographic location.

An alternative definition of “water security” refers to a water system’s ability to prevent and recover from physical security threats such as water contamination from chemical, biological and radiological agents.

Because of varied definitions of “water security,” and to highlight the relationship to public health, we use “water insecurity” as our reference term. Preventable direct health outcomes of water insecurity include water-borne illnesses, exposure to contaminants and toxins, dehydration and malnutrition. Indirect outcomes include emotional distress, depression and anxiety. Understanding water quality and quantity needs and inequities in access to safe water in Oregon is a prerequisite to developing community-specific and culturally-relevant water security policy solutions to help communities build adaptive capacity, strengthen resiliency and protect the health of all people in Oregon.

The Millennium Development Goal Report (MDG) of 2015 estimated that “663 million people worldwide use unimproved drinking water sources, including unprotected wells, springs and surface water, while 2.4 billion use unimproved sanitation [2].” Water security initiatives have focused heavily on developing countries to meet, by 2015, the Millennium Development Goal of halving those without sustainable access to safe drinking water and basic sanitation [3]. Since 1990, there has been some success in meeting this goal and significant increases in improved access to drinking water and sanitation. However, marginalized groups and rural communities still have inequitable access to piped water [2].

United Nations Water, the Global Water Partnership, and the World Economic Forum are a few of the international organizations focused on global water security efforts. In 2010, the United Nations General Assembly and the Human Rights Council recognized the human right to water as part of international law and the human right to sanitation followed as a distinct right in 2015 [4]. Under these rights, all people should have physical and affordable access to enough safe water and sanitation for personal and domestic use [4]. These rights, however, do not entitle people to free water, unlimited use or a household connection. Therefore, policy development is critical to supply affordable water and sanitation services, enough water for personal and domestic uses and water and sanitation access within or near the household [3].

In the United States (U.S.), water security is critical to protect public health. Drinking water contamination disrupts water access leading to poor health outcomes from exposure and a sense of concern among communities at risk [5]. Figure 1 shows the number (928) of waterborne disease outbreaks associated with drinking water reported in the U.S. during 1971–2014, by year and cause of disease or etiology. Two or more waterborne illness cases must be linked epidemiologically to be considered an outbreak [5]. Single cases, while not investigated, contribute to the background rate of infection from waterborne disease. There have been significant increases in reporting of bacterial outbreaks of *Legionella* in the last decade.

In more recent data, Figure 2 shows the percentage of drinking water-associated outbreaks reported in the US during 2013–2014, by chief illness and cause. There were 42 such outbreaks during this 2-year period. Acute gastrointestinal illness was associated with 41% of the 42 outbreaks, acute respiratory illness with 57% of the outbreaks and other illnesses accounted for 2% of the outbreaks. All reported outbreaks accounted for at least 1006 cases of illness, 124 hospitalizations and 13 deaths [5]. *Legionella* accounted for all acute respiratory illness and the causes of acute gastrointestinal illness outbreaks included exposures to *Cryptosporidium, Giardia, Clostridium, E. coli, Campylobacter,* norovirus, cyanotoxins, nitrite and 4-methylcyclohexanemethanol (MCHM) [5].

Two of the 42 waterborne disease outbreaks occurred in Oregon, affecting individuals served by community water systems. In June 2013, Oregon reported 119 cases of *Cryptosporidium* from a lake or reservoir resulting in two hospitalizations. In the following year, four cases of *Legionella* associated with well water resulted in four hospitalizations and one death [5]. The number of outbreaks and illnesses is likely underreported; symptoms are usually mild and resolve quickly. This reveals significant public health concerns for those in Oregon who lack access to clean and safe community water supplies for drinking and food preparation.

The Oregon Health Authority (OHA) administers the Safe Drinking Water Act (SDWA) and related state laws to ensure that communities on public water systems have access to drinking water that meets regulatory standards. Roughly 80% of Oregonians get their drinking water from public water systems. Oregon regulates public water systems that have four or more service connections or serve 10 or more people per day [7]. Through a partnership between OHA and the Oregon Department of Environmental Quality (DEQ), OHA’s Drinking Water Protection Program helps protect Oregon public water system sources (streams, lakes and aquifers) from contamination. Many rural community water systems have limited capacity to withstand drought, degrading infrastructure or declining resources to support system safety and integrity [8].

OHA estimates approximately 20% of Oregonians use private domestic wells as their primary source of potable water. The private well-owner has the responsibility to maintain their well and ensure the water is safe to drink. Rural or remote residents served by private wells with low means to assure adequate supply and quality are disproportionately affected by water insecurity. This poses a public health challenge to provide well-testing resources and to educate private well owners on the importance of wellhead stewardship, well maintenance, water testing and treatment if contaminants are present at elevated levels of concern. OHA’s Domestic Well Safety Program encourages well water protection by increasing well-owner capacity to evaluate and manage contamination risks.

Water insecurity disproportionately affects populations experiencing homelessness in Oregon. In 2017, a point-in-time count (a count of homeless people on a single night) estimated that 13,953 people were experiencing homelessness in Oregon, a 6% increase since 2015 [9]. Many people experiencing homelessness rely on public facilities for sanitation and hygiene. A research team at Portland State University surveyed 550 homeless people about where they access water for sanitation and hygiene. The team found that 55% use public bathrooms at the Central Library, City Hall and the mall, 33% used freestanding public toilets on downtown sidewalks and 32% used shelter rest rooms [10]. Forty percent of those surveyed reported experiencing medical problems related to lack of hygiene including staph infections, Methicillin-resistant *Staphylococcus aureus* (MRSA), endocarditis and urinary tract infections, observations which are consistent with chronic lack of sanitation and hygiene [10]. The survey results present an opportunity to further identify water access needs and public health impacts among those experiencing homelessness in all Oregon counties.

Water insecurity impacts households with poor plumbing, with wells vulnerable to flooding or with dependence on bottled water for daily use. Water insecurity also impacts those communities suffering from extended drought and those served by coastal aquifers vulnerable to sea level rise or saltwater intrusion. Oregon currently has no public health-focused water insecurity program, but partner programs within OHA such as Drinking Water Services, Domestic Well Safety Program (DWSP), Climate and Health Program and the Acute and Communicable Disease Program (ACDP) are working in this area. Despite current efforts, there is limited understanding of water insecurity risks and interventions to mitigate those risks. Public health-focused water insecurity programs, policies and practices could foster community resilience in the face of climate change, droughts, floods, wildfires, earthquakes and other natural disasters and communicable diseases related to water insecurity.

### Objectives

The objective of the systematic literature review was to understand the existing evidence base for developing public health policies, programs and surveillance strategies (collectively referred to as “interventions”) relating to water insecurity in Oregon. The following questions guided our literature review:(1)What are the existing evidence-based public health-focused water insecurity interventions?(2)How would one evaluate effectiveness?(3)What is the evidence showing the interventions have successfully mitigated risk?

## 2. Methods

We conducted a systematic literature search to collect existing documentation of evidence-based public health-focused water insecurity interventions. We followed Preferred Reporting Items for Systematic Reviews and Meta-Analyses (PRISMA) guidelines to document our literature review process [11]. Our aim was to document an evidence base for multiple interventions based on a mixture of qualitative and quantitative research. Therefore, we could feasibly produce neither an aggregate measure of effect nor a critical appraisal of studies.

### 2.1. Search Strategy and Screening Criteria

We looked for sources discussing water insecurity at the individual, household, community, state and national level with a public health focus. While we primarily targeted United States-based sources, we also reviewed international literature to understand the current global water insecurity political landscape and to assess the relevance and applicability of global water insecurity interventions in Oregon. We conducted the review using the ScienceDirect, PAIS Index and Nexis Uni databases to search for peer-reviewed journal articles and legislative documents. We also hand-searched the Water Security Journal from the ScienceDirect database and reference lists of relevant articles. Based upon initial search results, key informant interviews and an internet search of “university water initiatives,” we searched gray literature finding 15 separate agency and university water initiatives, their publication lists and research projects. We hand-searched the following institutions: California Water Boards, California Environmental Protection Agency’s Office of Environmental Health Hazard Assessment, Columbia University, the Environmental Law Institute, Harvard University, Northwestern University, Stanford University, Texas A & M University, the UCLA Luskin School of Public Affairs, the University of California Water (UC Water), the University of North Carolina, the University of Oklahoma, the University of Minnesota, West Virginia University and the World Economic Forum.

The search strategy used a combination of keywords including “water security,” “water insecurity,” “water scarcity,” “water access” “water stress,” “public health” and “surveillance.” Appendix A, Table A1 describes the complete strategy for the Water Security Journal in the ScienceDirect database. We assessed article abstracts for relevance before retrieving full texts. We used The George Washington University Himmel Health Sciences Library and the State Library of Oregon to obtain access to full-length articles.

The screening process used the following eligibility criteria:(1)Results for all years.(2)Results in the English language only.(3)Water insecurity-related bills, legislation, policies, journal articles, and agency publications, projects and initiatives.(4)Sources discussing measures to evaluate or identify water insecurity.(5)Sources discussing metrics for surveillance of water insecurity and public health impacts.(6)Sources discussing the application of indicators to identify water insecurity.(7)Sources documenting implementation processes, data collection methods or evaluation protocols for gathering evidence of effectiveness.(8)Sources currently in the implementation or evaluation phase that have documented implementation and evaluation strategies.

### 2.2. Data Extraction and Quality Assessment

To conduct data extraction, we made a distinction between systematic review protocols and systematic mapping protocols. The Collaboration for Environmental Evidence (CEE) provides a guideline for authors who seek to document environmental evidence [12]. During the data collection process, we used coding as suggested in the CEE’s systematic mapping guidelines to uniformly describe each intervention (see Table 1). We assessed the risk of bias in each study by determining whether the authors took measures to minimize selection bias and whether the study samples were prone to recall, nonresponsive, volunteer or response bias.

## 3. Results

Since 1998, literature has cited at least 25 different definitions of water security [4,13]. The Jepson et al. household water insecurity review [13] classifies these varied definitions into 4 interdisciplinary themes: (1) human needs and development; (2) ecological sustainability; (3) geopolitics and international relations; (4) vulnerability, adaptation and risk to global change. The Jepson review further distinguishes by level (e.g., individual, household, community, country, global) and water security frame (e.g., humanitarian, vulnerability, ecosystem sustainability, geopolitics) [13]. There are over 400 peer-reviewed water security publications in the social, natural and medical sciences field, with over half appearing in the last 5 years [14]. The Cook and Bakker [15] water security literature review revealed 95 results using the search term “water security” in the Web of Science database. Of the 95 articles, the majority were water resources, environmental studies and engineering focused, while fewer than 10 articles focused on public health. Search results did not identify the number of cited definitions for water insecurity.

### 3.1. Study Selection

Appendix A, Table A2 shows the Literature Search Log of all searches, including the database or site, keywords used, search results and relevant results. Figure 3 shows the PRISMA Flowchart for the literature search. We assessed 2323 articles and gray literature items for relevance. After completing the screening phase, eleven results passed criteria for inclusion: eight full-text articles and three gray literature results. The articles that passed screening were found either through agency publication lists or the Elsevier Water Security Journal, while the gray literature results were found through follow-up searches originating from Nexis Uni legislative results. Out of the 15 separate agency searches, The University of North Carolina and Texas A&M University yielded results that we could include in the screening process. We grouped all results meeting criteria for inclusion into three categories which include policy, surveillance and indicators.

#### 3.1.1. Policy

##### California Human Right to Water Assembly Bill 685

The literature search found one state policy meeting the search criteria and study objectives. The 2012 California Human Right to Water Assembly Bill (AB) 685 recognizes “every human being has the right to safe, clean, affordable, and accessible water adequate for human consumption, cooking, and sanitary purposes [16].” AB 685 requires that all relevant state agencies consider the policy when developing and implementing related policies and programs. Indicators and evaluation methods to monitor progress and assess achievements of the policy are still under development in partnership with the University of California Berkeley (UC Berkeley). UC Berkeley School of Law released a document called “The Human Right to Water Bill in California: An Implementation Framework for State Agencies” in May 2013, which offers a framework defining how and when state agencies should consider the human right to water [17].

#### 3.1.2. Surveillance

There were seven surveillance studies meeting criteria for inclusion addressing water insecurity at the household, community, regional and or state level. For this review, we define surveillance as public health data collection and analysis to identify water access needs, to identify populations experiencing water insecurity and to note existing public health inequities.

##### Household Water Insecurity Studies

Two studies assessed water insecurity at the household level, with a focus on poor communities in both international and US-based communities. The Global Household Water Insecurity Study at Northwestern University, launched in 2017, identifies households with high, medium and low water insecurity using the Household Water Insecurity Experiences (HWISE) Scale [18]. It is currently implemented as a cross-culturally validated tool in 23 low-and middle-income countries [18]. The HWISE Scale is designed to assess risks of adverse outcomes associated with household water insecurity (HWI), to target scarce resources and to measure impacts of interventions and policies on HWI [18]. Approximately 250 participants were randomly selected per site to complete the household water insecurity surveys. The survey included 32 interview questions about water insecurity (for example, socio-demography, water quality, quantity, accessibility, reliability and utility, food insecurity, perceived stress and infant feeding). While the project offers a comprehensive guidebook and rationale for conducting a HWISE study, the project is still underway with no results to assess the effectiveness of the intervention. Ongoing data collection is viewable on the study site until a final scale and study is published [18].

The Jepson household water security study [19] took place in colonias, which are low-income communities along the US-Mexico border. The study focused on Hidalgo County, Texas, where the researchers conducted 71 household surveys over four weeks in 2012 [19]. The study used key terms such as “water security” and “household water security”, which encompass three dimensions: water access, water quality acceptability and water affect [19]. Water affect is the “emotional, cultural, and subjective experiences of water” [19]. The aim was to gather water security perspectives of colonias residents using qualitative research and experiential surveys. The study developed a scalogram, which is a cumulative scale incorporating each of the three dimensions of household water security mentioned. At the time of the study, all households surveyed had a water service connection. Researchers selected 11 colonias communities through random sampling using the Texas state colonia classification system [19]. The system classified six border counties according to infrastructure-based assessments of high, moderate, low and unknown health risk [19]. The scale scores to classified households as: (1) Water Secure; (2) Marginally Water Secure; (3) Marginally Water Insecure; (4) Water Insecure. The study found that “only 10% are water secure, 35% are marginally water secure, 31% are marginally water insecure, and 24% are water insecure [19].” This study offers the perspective of marginalized communities who are experiencing water insecurity.

##### Socioeconomic Status, Race/Ethnicity, and Risk of Unsafe Drinking Water Study

The Switzer and Teodoro study analyzed direct and indirect relationships between racial or ethnic populations, socioeconomic status (SES) and Safe Drinking Water Act (SDWA) compliance [20]. The study compared two regression models, one noninteractive model assessing the effect of race and ethnicity on SDWA compliance and one interactive model including SES as conditional variable. The comparison strengthened the evidence for a conditional relationship between SES and race and ethnicity. Results indicate that SES is proportional to compliance and ethnicity (Hispanic) is inversely proportional to compliance. There was no significant relationship between race (black) and compliance. When Switzer and Teodoro included SES as an interactive term (descriptor of the population), the effects of increasing community race and ethnicity on drinking water violations declined as SES increased. Low SES communities with racial and ethnic minorities (Hispanic and black) face greater risk of unsafe drinking water [20].

##### Municipal Water Service Access Study

The Gibson et al. study [21] used property tax data in Wake County, North Carolina to quantify the percentage of residences with community water service in each census block. The study provides the first systematic identification of communities on the fringes of towns in Wake County, North Carolina, lacking access to municipal water service. Researchers conducted the study to test the hypothesis that race may play a role in access to community water service in areas at the fringes of North Carolina cities [21]. To determine if race was a significant predictor of water service access in census blocks at the fringes of North Carolina towns, researchers conducted logistic regression analysis [21]. The study showed that increases in the African American population proportion within a census block correlated with an increase in the odds of exclusion from municipal water service [21].

##### Water Service Reliability Study

The Pierce and Jimenez study [22] used 2011 housing data to find disparities in water service reliability among mobile home communities. Researchers hypothesized that households receiving water service from small systems (defined in this study as systems not regulated by Safe Drinking Water Act, or those systems that serve fewer than 15 connections or 25 people) experience more interruptions in service than larger systems. Further, small potentially unregulated systems are less likely to be properly maintained and thus are prone to gaps in service. The study analyzed water reliability across three housing types: standalone mobile homes, mobile homes in park communities and all other housing units. Researchers administered surveys with questions on household socioeconomics, tenancy arrangements, housing quality, costs, relocation behaviors and location [22]. Results showed that mobile home park residents experience nearly twice the number of service gaps of residents of standalone mobile homes and nearly three times as many as residents in all other housing unit types. Units receiving water from a small system had 3 times higher odds of experiencing a water shutoff than other units [22]. More research is needed to illuminate the underlying reasons for these differences; however, it is likely that gains in water security are more feasible in mobile home parks than among standalone mobile homes. Residing in a rural area within a metropolitan statistical area (MSA) was more strongly associated with unreliable water service than location in a non-MSA rural area.

##### Water Scarcity Variability and Exceedance/Compliance Mapping Tools

Two studies conducted water insecurity surveillance through the development and use of mapping tools. The Mekonnen and Hoekstra study [23] measured global water scarcity monthly with remote sensing at high resolution (30 × 30 arc min). Water scarcity is the ratio of water consumption over water availability. Researchers adopted an environmental flow standard that 80% of natural runoff is needed to meet environmental needs, leaving 20% which can be considered as water available for human use without affecting the integrity of downstream water-dependent ecosystems and livelihoods [23]. Results show that 4 billion people worldwide experience severe water scarcity during part of the year and 1.8 to 2.9 billion people experience severe water scarcity for 4 to 6 months each year [23].

The California State Water Resources Control Board recently published the Human Right to Water Portal [24]. The Portal includes a data mapping resource showing public water system compliance and violations information in California [25]. The tool uses available data to identify the water system number and name, the regulating agency, the county, the service connections, city, zip code, compliance status and the violation details (type of contamination).

#### 3.1.3. Indicators

##### Water Poverty Index Studies

One of the most widely used indicators relating to water and human development is the Water Poverty Index (WPI), first applied at the community level by Sullivan, et al. [26]. The WPI integrates information about local water resources, access to water, capacity to manage water, uses of water and ecological integrity to determine risks and priorities of water conditions at the community level. The WPI takes a value of 0–100, with 100 being the best situation (low level of water poverty) and 0 being the worst (high level of water poverty).

Sullivan, et al. [26] demonstrated the value of the WPI as applied to twelve international community pilot sites in South Africa, Tanzania and Sri Lanka. This study identified WPI index values for each community through the administration of 1521 household surveys. Results quantified strengths and weaknesses related to water resources for each pilot site. The WPI provided a cogent, transparent way to communicate the complexities of local water issues and their impacts on local communities.

Korc and Ford [27] applied the WPI model in 131 households along the border colonias of west Texas. The WPI indicators include:Resources (capacity of water systems and water quality of suppliers in a colonia),Access (access to drinking water and sanitation, and institutional or technical capacity of water suppliers in a colonia),Capacity (cost of water, household annual income and drinking water tank maintenance in a colonia) and,Environmental (septic tank certification and septic tank maintenance in a colonia) [27].

To create a colonia-level WPI, the study matched a water, sanitation and safety dataset from a 2010 Texas Department of State Health Services community-based survey on colonias with water supplier compliance and enforcement information. Researchers combined the WPI components using a weighted average method, with the weightings indicating the importance of a particular WPI component [27]. Results identified the neediest of the studied colonias and showed that improvements in any of the 4 indicators would benefit that community. Results also identified specific needed improvements for the other colonias studied, allowing for more focused community resources [27].

##### A Framework for Evaluating California’s Human Right to Water

On January 3, 2019, the California Environmental Protection Agency’s Office of Environmental Health Hazard Assessment announced the release of a draft of *A Framework and Tool for Evaluating California’s Progress in Achieving the Human Right to Water* [28]. The Framework offers a systematic approach for evaluating California’s AB 685 law, using 13 indicators which quantify drinking water quality, accessibility and affordability across the state’s community water systems [28]. Individual systems get an overall score in each indicator area. Implementation and evaluation of AB 685 is an ongoing process, with important considerations such as changes in policy and methods to assess water access, quality, and affordability [28]. 

### 3.2. Risk of Bias and Critical Appraisal

Since our review does not seek studies with intervention and outcome measures of a clinical nature, and includes a variety of results (e.g., policy, indicator and surveillance tools), we cannot determine risk of bias across studies. We critically appraised the descriptive [18,19,26,27] and correlational studies [20,21,22] using a modified Critical Appraisal Skills Programme (CASP) checklist [29]. Four sources are not applicable to the critical appraisal framework [16,23,25,28]. Responses in Table 2 indicates that all descriptive and correlational studies present strong internal validity, detailed results and relevant and valuable research.

## 4. Discussion

### 4.1. Summary of Evidence and Potential Utility in Oregon

Our literature catalog Appendix A, Table A3 shows that current public health-related water insecurity interventions are limited and mainly consist of surveillance practices (for example, surveys) with some water insecurity mapping and indicator strategies. There are few water insecurity policies that seek to promote public health and health equity. Study findings may be difficult to find due to varied definitions and uses of water insecurity and water security. However, research and development are most likely limited by the relatively recent emergence of water insecurity as a field of study. Knowledge is still emerging on how to quantify water insecurity to assess community needs and develop effective interventions to mitigate risk. In Table 3, we present intervention feasibility (ability/likelihood to complete the intervention successfully in Oregon) and relevance (intervention importance or significance for Oregon).

Given the United Nations’ and California’s Human Right to Water Laws, we see increasing global, national and regional efforts to achieve access to safe, sufficient and affordable water for all people. With California’s *Framework and Tool for Evaluating California’s Progress in Achieving the Human Right to Water* [28], policymakers and public health professionals have a model for tracking progress towards state water security goals. While the Framework was designed to assess progress of California’s AB 685, it can be adapted in other states to evaluate existing water policy efforts. Since California is the only U.S. state thus far to declare a human right to water, it is too early to determine the full impact of the Framework indicators.

The two household water insecurity studies [18,19] measure multiple aspects of water insecurity to assess what is experienced at the household level. Resulting information offers policymakers well-rounded data to apply an integrated approach to designing water insecurity interventions. The Household Water Insecurity Experiences (HWISE) Scale equips researchers with a tool to identify the magnitude of household water insecurity, the associated health risks and effects on wellbeing, changes over time and intervention effectiveness [18]. The HWISE study offers a complete guidebook for research teams, however, the large-scale study design has high resource and time demands not suitable for projects with significant budget and time constraints.

The colonias experiential survey and scales integrate the point of view of marginalized communities experiencing water insecurity, supplying qualitative data and quantitative analysis to classify varied levels of water insecurity and water security. The three scalograms (water access, water quality acceptability and water affect) together offer information on multiple dimensions of water insecurity and capture psycho-social dimensions (for example, emotional distress) often overlooked in water insecurity measurements [19]. The scale is reproducible, since survey questions are general and prompt the participant to share their own unique experience. The scale is most applicable to low-income and unincorporated peri-urban and rural communities connected to water services, but the scale can be adapted to assess communities not connected to water services as well.

For instance, water insecurity scalograms can be applied to rural communities with individual surface water domestic withdrawals in Oregon (for example, Siltcoos Lake). The risk of selection bias for both studies is low since researchers used random sampling to identify participants. Both studies used primary and secondary data offering a mix of qualitative and quantitative information.

Two studies investigated the relationship between race and ethnicity and water insecurity [20,21]. Switzer and Teodoro [20] revealed a statistically significant negative relationship between race/ethnicity and Safe Drinking Water Act (SDWA) compliance in communities with exceptionally low socioeconomic status (SES). The study controlled for a possible confounder (SES) by comparing a non-interactive regression model (race and ethnicity and SDWA compliance) to an interactive model (race and ethnicity and SDWA with SES as a confounding variable). Since this is the only study found which assesses these interactions, we cannot assume that there will be a negative relationship between race and ethnicity and SDWA compliance in all low SES communities. Oregon, and other states interested in this issue, would have to determine if these relationships reflect what is experienced in local communities. The Gibson et al. study [21] indicated that increases in the African American population proportion within a census block correlated with an increase in the odds of exclusion from municipal water service. Both studies’ use of secondary data (for example, census records and SDWA data) present a financially feasible approach to identifying water insecurity disparities. Results present an opportunity to identify and investigate further the disparities in access to water in similar communities in Oregon.

The Pierce and Jimenez study [22] compared water service reliability of households on small water systems not regulated by the federal Safe Drinking Water Act with households on larger systems. Results show that mobile home park residents are 2–3 times as likely as others to experience gaps in water service [22]. Researchers used surveys and secondary data (for example, U.S. census and American Housing Survey data) to identify disparities in water service reliability; therefore, it is feasible to conduct a similar study in Oregon. However, current American Housing Survey data are limited to a combined (Portland, Beaverton and Vancouver) metropolitan area dataset so comparable data are needed. Conducting a similar study would provide baseline data to show the prevalence of water reliability in Oregon communities on small water systems.

The Mekonnen and Hoekstra [23] study on water scarcity variability mapping and the California Water Boards’ exceedance/compliance mapping tool offer water insecurity surveillance tools for both small and large-scale tracking. While the study aimed to reveal large-scale impact, researchers can use remote sensing to assess water scarcity regionally. Researchers interested in modeling water scarcity can use data from domestic water supply institutes and the United States Geological Survey (USGS) Water Use Data for the Nation [30] database to measure water consumption of domestic water supply. Surface runoff modeling at catchment or grid level can measure water availability. The California Water Boards’ exceedance/compliance mapping tool [25] is the first of three drinking water mapping tools available to track progress toward the human right to water. Once complete, the two additional maps will provide methods for tracking water affordability and water accessibility and other important dimensions of water insecurity. With existing Oregon data (for example, water system names and numbers, regulating agencies, service connections, compliance status and violations) it is possible to create a similar mapping tool of public water systems. One limitation to California’s exceedance/compliance mapping tool is the exclusion of private water systems, small state-regulated water systems and private wells. Since California’s exceedance/compliance tool only maps public water systems, Oregon would need to assess alternatives to tracking data on other water systems of interest in the state.

The Korc and Ford [27] and Sullivan et al. [26] studies present the Water Poverty Index (WPI) as a water management tool that considers many aspects of water management including data on water resources, access, use, socio-economic capacity and water quality [31]. Although the WPI calculation process requires multiple data sets, the multi-step process is simple with examples of the standard WPI equation available to researchers [26]. Selecting indicators from available data will capture a more informed picture of the five components of the index. The Korc and Ford [27] study used existing datasets from three sources: the 2010 Texas Department of State Health Services (TDSHS) community-based survey, the Texas Commission on Environmental Quality (TCEP) water supplier inspections and the Safe Drinking Water Information System (SDWIS). Researchers can still derive approximate WPI results with missing data, however, this may reduce the comparability between locations [26].

### 4.2. Review Limitations

Currently, there is no database or centralized source compiling a list of agencies and academic entities with water insecurity initiatives. Since we hand-searched state-and university-based water initiatives, it was difficult to determine if we found all existing initiatives and risk of search bias exists. Much of current literature used “water security” as a key term, yielding more relevant results than “water insecurity”. Most results excluded from the review were conceptual studies discussing varied definitions of water security. This presented challenges with identifying proper search terms and sources specifically addressing a public health perspective and quantifiable solutions. While “water insecurity”, “water security”, “the human right to water”, “household water insecurity”, “water poverty index”, “water scarcity”, “water stress”, “water service reliability” and “water access” surfaced as key terms in our literature results, it is possible that other key terms are used to encompass water insecurity. These challenges suggest that water insecurity is still an emerging topic with limited publications.

### 4.3. Key Informant Interviews

At various stages of our systematic review process we interviewed key informants actively involved in solving today’s water insecurity issues in California, Washington and Oregon. Discussions in Oregon focused on natural resources agencies, since activities within the public health system (Oregon Health Authority programs regulating public water systems and carrying out education and outreach to private domestic well owners) are already known to the authors. Key informant interviews contributed to the robustness of key words and search terms allowing us to refine our review, expand our understanding of relevant policies and begin identifying potential public health-focused water insecurity interventions. Below are the key activities discussed during the interviews.

California’s Human Right to Water Law followed mobilization of multiple communities and nongovernmental organizations focused on water justice. These communities recognized inequities in access to safe drinking water. California is assessing water security needs, setting goals, developing resources, improving water affordability and tracking progress. Washington State is pursuing water insecurity interventions through community education events, drought risk assessment, water loss detection and utility rate capping. Washington’s activities look at water insecurity outside of the regulatory context and aim to increase awareness of the value of water and importance of water conservation. “Water insecurity” is not a widely used term in California or Washington, but both states are developing interventions to meet the unique water needs of their communities.

There are recent and ongoing efforts among Oregon natural resource agencies, led by the Oregon Water Resources Department (OWRD), to develop an Integrated Water Resources Strategy, gain a better understanding of the quantity and quality of water in aquifers, prepare for droughts, engage local communities in place-based planning and develop a 100-year vision for water in Oregon. To capture groundwater-level data and climatic and seasonal impacts on aquifers, OWRD continues to develop observation wells throughout the state and in basin study sites. OWRD has completed basin studies in the Deschutes, Willamette and Klamath basins to understand the relationship between and availability of groundwater and surface water. These studies produce information about the geology of the basins and the volume of groundwater recharge, discharge and storage [32]. In response to the 2015 drought, partner agencies of Oregon’s Integrated Water Resources Strategy conducted a survey across the state asking participants about their drought response and what actions should be taken to be better prepared for future droughts [32].

Oregon Department of Environmental Quality’s (DEQ) drinking water source protection program [33] assists public water systems and communities with protecting their sources of drinking water (streams, lakes and aquifers) from contamination. DEQ conducts and updates source water assessments for public water systems and develops surface water and groundwater resource guides to provide technical assistance, funding information and other resources to public water systems in Oregon.

### 4.4. Recommendations

The following recommendations are based on key informant interview findings and current water insecurity literature, policies and practices (2003–2019) that aim to identify public health inequities in access to safe, accessible and affordable water. Our goal is to inform public health agencies, water resources management agencies and policymakers about water insecurity inequities and interventions to make the most informed decisions to mitigate risk.

#### 4.4.1. Characterize Populations Experiencing Inequities in Access to Water

Prior to identifying water insecurity needs, state agencies must recognize which communities are experiencing water access disparities. Agencies should study census and property tax data to identify, describe and quantify inequities in access to water. This will help us understand racial and ethnic disparities and water access needs among marginalized communities (for example, communities with no or disconnected water service and communities along the fringes of municipal water service lines). Mapping Oregon water systems as done in California’s Exceedance/Compliance mapping tool [25], and considering inclusion of water system advisories, will help us understand water availability and water quality among populations served by these systems. For instance, we can map communities that have open or recurring water advisories. Both approaches will focus risk mitigation activities, such as improving well construction, updating infrastructure, educating private well owners on well maintenance and groundwater stewardship and educating communities on drought awareness and water conservation.

#### 4.4.2. Identify Water Insecurity Needs

Assessing water insecurity experiences and needs at the household level using the Household Water Insecurity (HWI) approach [19] will find water-insecure households and communities and help us understand the impacts water insecurity has on daily life, health perceptions and psychosocial well-being. Researchers can distribute and analyze HWI surveys using the Community Assessment for Public Health Emergency Response (CASPER) technique, a quantitative and evidence-based needs assessment approach [34]. Although this approach is resource-heavy, data acquired from the first recommendation will reveal target communities for its application and baseline data to support funding requests. Qualitative data from the HWI surveys paired with existing quantitative data (for example, water quality data, service interruptions and water-borne illness rates) will produce valuable information about the unobserved experiences of communities in mobile home parks, communities with individual surface water domestic withdrawals and communities with private wells.

The Water Poverty Index (WPI) identifies more than one influencing factor (that is, Resource, Access, Capacity, Use and Environment) of water insecurity, which provides an interdisciplinary picture of the water situation in a given region [31]. The WPI, and its integration of existing data and the physical, social, economic and environmental issues, can be an effective and feasible tool for water managers in Oregon to develop sustainable water resources. Application of the WPI in Oregon communities will require resources to collect data, analyze multiple data sets and calculate the WPI. A research team could conduct a household-level study, such as Jepson’s HWI [19], in conjunction with a WPI study or use data from a previously conducted household survey.

Oregon Housing and Community Services (OHCS) developed a Manufactured Home Park Directory map, which includes the locations of all active manufactured homes (including mobile homes), marked census track boundaries, county names, park size or number of manufactured homes and owner contact information [35]. To assess water service reliability in manufactured/mobile home parks, American Housing Survey (AHS) questions [22] can integrate into HWI survey studies to capture physical housing quality attributes, housing costs and tenancy arrangements, relocation behaviors and water cutoff or service interruption rates. This approach would quantify the needs of low socioeconomic status (SES) and marginalized communities, identifying inequities in access and health impacts of water insecurity. Currently, AHS data in Oregon are limited to the Portland-Beaverton-Vancouver metropolitan areas, however it is possible to match some manufactured/mobile home parks in the OHCS Manufactured Home Park Directory map with mobile home parks and corresponding water connections through the Drinking Water Data Online inventory [36]. There may be opportunities to work with partners to fill critical data gaps and provide more transparency to water insecurity in marginalized communities.

#### 4.4.3. Track Water Scarcity Variability

Oregon can implement strategies to better understand how climate change and seasonal variability impacts water availability and drought risk by applying Mekonnen and Hoekstra’s water scarcity modeling approach. Researchers can measure water scarcity as the ratio of water consumption over water availability. The U.S. Geological Survey provides water-use data per county, including domestic self-supplied groundwater and surface water withdrawals measured in Mgal/d (million gallons per day). Water availability is measurable through surface runoff modeling at the catchment or grid level [23]. Oregon should consider updating or developing surface runoff water modeling and aquifer modeling at the catchment or grid level. Water scarcity mapping results will support Oregon’s Integrated Water Resources Strategy by monitoring and quantifying water consumption and availability in relation to climate change and drought patterns. Results can also aid the mobilization of place-based water planning efforts in regions found to be experiencing water scarcity.

#### 4.4.4. Pursue the Human Right to Water

Adopt a state policy, such as the human right to water, that would encourage an interdisciplinary collaboration between public health agencies, water resources management agencies, academia and policymakers to establish accessible, safe and affordable water as a priority in future policy decisions. Resulting data from the preceding recommendations (for example, water insecurity experiences, water access and demographics, water quality and availability mapping and service reliability) will inform future policy decisions.

## 5. Conclusions

Water insecurity as an emerging field opens many opportunities to raise awareness of the evolving and confounding effects to public health, as well as understand how to mitigate and manage risks of human and ecologic impacts. The current evidence base for making policy- and program-development decisions is limited, but there are feasible and relevant intervention options to begin assessing water insecurity needs in Oregon. Work around California’s Human Right to Water and related mapping tool, and additional applications of the Household Water Insecurity Survey by researchers over the next few years will help to grow the evidence base. The United Nations and California establishment of the human right to water suggests policymakers and public health professionals are placing a greater importance on environmental justice and water insecurity internationally and in the US. Meetings with US-based key informants revealed that water insecurity policy and intervention development is underway, but each state interviewed (OR, WA, CA) is pursuing its own approach. While approaches may differ and cater to specific state-based needs, there is great opportunity to learn about varied approaches through continued collaboration. By using mixed methods to collect information at system, community, regional and statewide scales Oregon will be better prepared to develop water insecurity policy solutions that are community-specific and culturally relevant.

## Figures and Tables

**Figure 1 ijerph-17-01122-f001:**
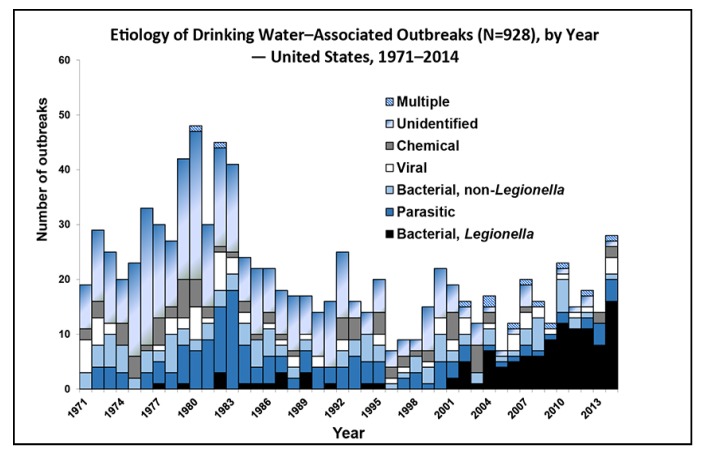
Etiology of 928 drinking water—associated outbreaks, by year-United States, 1971–2014 [6].

**Figure 2 ijerph-17-01122-f002:**
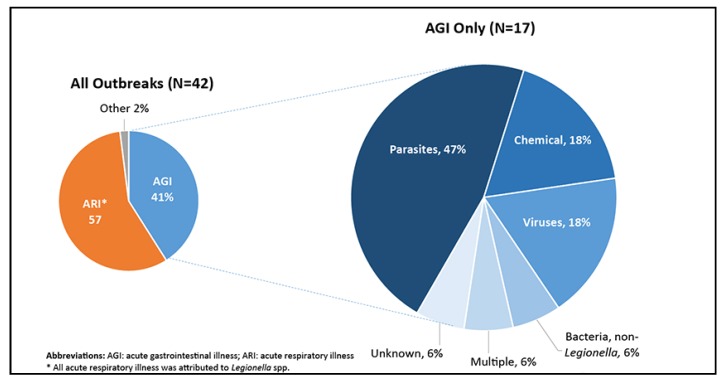
Chief Illness Reported for all Drinking Water Outbreaks, and Etiologies in Outbreaks of Acute Gastrointestinal Illness (AGI), 2013–2014 [6].

**Figure 3 ijerph-17-01122-f003:**
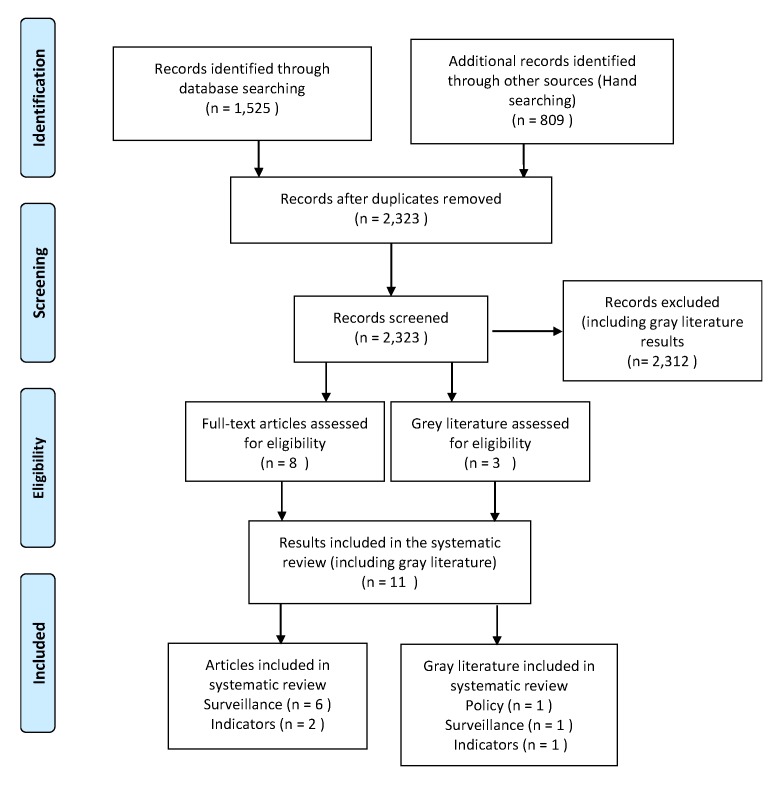
Prisma Flowchart of Review Results.

**Table 1 ijerph-17-01122-t001:** Study Coding Variables.

Coding Variable	Information
Source	Title, author(s), date
Publication type	Academic journal, book, conference paper or thesis
Study location	Name of country, state, region, or community
Study funding	Name source, or indicate none specified, could not locate
Data collection instruments	e.g., Primary or secondary data e.g., Quantitative or qualitative
Study Design	e.g., Observational, survey
Population	e.g., Low socioeconomic status communities, etc.
Intervention	e.g., Policy, surveillance, or indicator
Sampling strategy	e.g., None specified, randomized, or systematic
Length of study	e.g., Number of days, weeks, months, years or time-period over which the study was undertaken
Feasibility	Ability/likelihood to complete the intervention successfully in Oregon
Relevance	Intervention importance or significance for Oregon

Notes: Reprinted from A methodology for systematic mapping in environmental sciences [12].

**Table 2 ijerph-17-01122-t002:** Critical Appraisal of Descriptive and Correlational Studies.

Source	Was There a Clear Statement of the Aims of the Research?	Is a Qualitative Methodology Appropriate?	Was the Quantitative Methodology Appropriate?	Was the Research Design Appropriate to Address the Aims of the Research?	Was the Recruitment Strategy Appropriate to the Aims of the Research?	Was the Data Collected in a Way that Addressed the Research Issue?	Has the Relationship between Researcher and Participants Been Adequately Considered?	Was the Relationship between A and B Explored? (Correlations Studies)	Have Ethical Issues been Taken into Consideration?	Was the Data Analysis Sufficiently Rigorous?	Is there a Clear Statement of Findings?	Is the Research Valuable?
Northwestern University	Y	Y	Y	Y	Y	Y	Y	NA	Y	Y	N/A	Y
Jepson et al.	Y	Y	Y	Y	Y	Y	Y	NA	Y	Y	Y	Y
Sullivan et al.	Y	Y	Y	Y	Y	Y	Y	NA	Y	Y	Y	Y
Korc and Ford	Y	Y	Y	Y	Y	Y	CT	NA	NA	Y	Y	Y
Switzer & Teodoro	Y	NA	Y	Y	NA	Y	NA	Y	NA	Y	Y	Y
Gibson et al.	Y	NA	Y	Y	NA	Y	NA	Y	NA	Y	Y	Y
Pierce and Jimenez	Y	NA	Y	Y	NA	Y	NA	Y	NA	Y	Y	Y

Notes: Key: Response options: Y = Yes; CT= Can’t tell; N = No; NA = Not applicable. Adapted from Critical Appraisal Skills Programme. CASP (Qualitative) Checklist [29].

**Table 3 ijerph-17-01122-t003:** Relevance and Feasibility of Interventions in Oregon.

Intervention Category	Reference Title/Author	Methods and Results	Risk of Bias or Limitations	Feasibility	Relevance
**Policy** State level	California Human Right to Water Assembly Bill (AB) 685 California Water Boards (2012)	The bill led to the development of an Office of Environmental Health Hazard Assessment framework draft for evaluating the human right to water progress	N/A	Oregon could legislatively recognize the human right to water in the future; however, water insecurity intervention and policy development can take place in the absence of such a law.	With the Oregon Clean Energy Jobs bill (HB2020) and climate change in the 2019 legislative session spotlight, it is worth discussing the push for a human right to water bill in future legislative sessions.
**Surveillance** Household Level	Household Water Insecurity Experiences (HWISE) Scale Northwestern University (2019)	Still being evaluated for effectiveness	Low risk of bias due to random sampling	HWISE could be conducted as a pilot study; however, feasibility in Oregon may be a challenge due to high resource and time demands.	Household water insecurity assessments are highly relevant in communities on small water systems/private systems in Oregon; however, smaller-scale household studies are more reproducible.
**Surveillance** Household Level	Jepson Measuring ‘no-win’ waterscapes: Experience-based scales and classification approaches to assess household water security in colonias on the US–Mexico border (2014)	Results: Survey identified 10% of households as ‘water secure,’ 35% as ‘marginally water secure,’ 31% ‘marginally water insecure,’ and 24% as ‘water insecure.’ Overall, only 45% are broadly water secure while 55% are water insecure.	Low risk of bias due to random sampling	The Household Water Insecurity (HWI) surveys/scalograms can integrate into the Community Assessment for Public Health Emergency Response (CASPER) framework to assess water insecurity needs, health status, and health perceptions during non-emergency and emergency situations. However, this approach is resource heavy and would need funding to staff a research team to conduct the study.	The HWI surveys are useful experiential measurement tools to assess water insecurity in Oregon communities on small water systems or private water systems such as: mobile home park units, standalone mobile homes, rural/unincorporated communities using private wells.
**Surveillance** Community Level Public water systems	Switzer & Teodoro Class, Race, Ethnicity, and Justice in Safe Drinking Water Compliance (2017)	Methods: Statistical regression isolated direct and interactive relationships between communities’ racial/ethnic populations, socioeconomic status (SES) (SES as a conditional variable), and Safe Drinking Water Act (SDWA) compliance. Results: Community racial/ethnic composition predicts drinking water quality. SES status strongly predicts SDWA violations. There is a statistically significant negative relationship between race/ethnicity and SDWA in communities with exceptionally low SES.	Risk of bias is low; however, we cannot assume that there is a negative relationship between race/ethnicity and SDWA in all low SES communities.	The study uses readily available census data and SDWA violations data; therefore, it does not require high money demands and staff resources.	This study is suitable for assessing water insecurity in low SES communities in Oregon from an environmental justice standpoint. It finds potential inequities in access to safe drinking water (SDWA violations), but water quality is just one aspect of water insecurity. To assess water insecurity and environmental justice holistically, inequities in water service reliability, affordability, and quantity are also variables to consider in race/ethnicity and low SES interactions.
**Surveillance** Community Level Public water systems	Gibson, DeFelice, Sebastian, & Leker Racial Disparities in Access to Community Water Supply Service in Wake County, North Carolina (2014)	(Used publicly available property tax data to quantify percentage of residences with municipal water service in each census block in Wake County) Every 10% increase in the African American population proportion within a census block increases the odds of exclusion from municipal water service by 3.8% (p < 0.05).	Low	The study uses readily available census data; therefore, it does not require money demands and staff resources.	This study is suitable for assessing water access and racial/ethnic disparities among marginalized communities just outside municipal water service lines.
**Surveillance** Community Level Small Water Systems (Pierce & Jimenez categorize networks serving fewer than 15 households as small systems)	Pierce & Jimenez Unreliable Water Access in U.S. Mobile Homes: Evidence from the American Housing Survey (2015)	Using housing data, the study finds disparities in water access, specifically water service reliability among mobile home communities.	The study aimed to assess water reliability only (water access shutoffs), but acknowledge that there are other relevant dimensions of household water access security (available volume, quality, and cost of water) Limitations: The American Housing Survey (AHS) data from 2011 groups three metropolitan areas together (Portland-Beaverton-Vancouver). And there currently is no data on mobile homes in other areas in Oregon.	The study used 2011 housing data and the American Housing Survey, both easily accessible. Currently, AHS data on Oregon manufactured/mobile homes is limited to the Portland and Beaverton metropolitan areas.	Although water service reliability is just one aspect of water insecurity (water access), the method would provide baseline data on service reliability in Oregon communities connected to small water systems (mobile home parks, apartments. etc.).
**Surveillance** (Mapping) State or Regional Level	Mekonnen Four Billion People Facing Severe Water Scarcity (2016)	The number of people facing severe water scarcity for at least 4 to 6 months per year is 1.8 to 2.9 billion.	Low	There are multiple ways to measure water scarcity, giving flexibility to the type of modeling we want to produce based on existing resources and limitations. To look at the water scarcity of domestic water supply, we can measure the consumption of water supply (United States Geological Survey data) over the water availability (surface runoff modeling/catchment grid level).	The water scarcity variability mapping is useful as a regional tracking tool and delivers water scarcity data that will inform health risks. The tool may be suitable for bridging water insecurity and the Oregon Health Authority (OHA) climate change program efforts.
**Surveillance** State level Public water systems	California Water Boards: Water Security Portal Exceedance/Compliance Status of Public Water Systems Map (2018)	The tool visually documents and maps exceedance/compliance data on California community water systems, offering a centralized and accessible database for community members, policy makes, and water regulators.	Limitations: The mapping tools in California currently excludes data for small regulated-water systems or private wells.	It is feasible to collate and map exceedance/compliance data on Oregon community water systems if data on water system names/numbers, regulating agencies, service connections, compliance status and violations are accessible/available.	California’s Exceedance/Compliance tool does not currently map water data on small water systems. If implemented in Oregon, this tool would represent more communities if it mapped data on small regulated-water systems/ private wells in addition to public water systems.
**Indicators** Community Level	Sullivan et al. The Water Poverty Index: Development and application at the community scale (2003)	Results: Study identified Water Poverty Index (WPI) values for each community.	Proxy data is used in place of missing data; however approximate results can be calculated when some data is missing.	Calculating a WPI is feasible due to its use of existing datasets: Safe Drinking Water Information System (SDWIS), US EPA National Pollutant Discharge Elimination System. The WPI provides a structured and easy to follow, 8-step WPI model	WPI is directed towards communities and poorer areas, incorporating environmental integrity and ecosystem water needs.
**Indicators** Community Level	Korc & Ford Application of Water Poverty Index in Border Colonia of west Texas (2013)	Methods: The WPI takes the value of 0-100, with 100 being the best situation (low level of water poverty) and 0 being the worst (high level of water poverty). Results: Varied results per location.	“Not all the components of water poverty were included in the index (e.g., water usage). The use of three datasets from different sources to calculate the WPI may affect the quality of the results (Korc & Ford, 2013).”	Same as above.	The WPI is relevant tool that can link household welfare with water availability in high poverty communities. WPI can integrate physical, social, economic and environmental information, which helps determine priorities associated with water in communities like the colonias.
**Indicators** State level	California Environmental Protection Agency, Office of Environmental Health Hazard Assessment	Methods: The framework presents 13 indicators of domestic water supply (water quality, water accessibility, water affordability) to monitor, track, and assess California’s human right to water. Results: the framework/indicators are still under development and have not yet been implemented by state water management agencies	N/A	Since the indicators focus on community water systems, they are reproducible in Oregon, but they are not designed to be adaptable to private systems. California is currently focusing on community system indicator development but plans to address private systems in the future.	The framework is relevant to community water systems; however, the indicators would not assess water insecurity in Oregon communities on private water systems

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
