# Peer review of "A Systematic Literature Review on Water Insecurity from an Oregon Public Health Perspective"

_ijerph, 2020, doi:10.3390/ijerph17031122_

Round 1

Reviewer 1 Report

The authors undertook a systematic and thorough literature review of the water insecurity literature with the aim of informing policy in Oregon. The review was formally structured and comprehensive and short summaries of each relevant publication are provided. These results are then applied to a specific context, the desire to assess and address water insecurity at the state le. The paper is interesting and generally reads well and I recommend that it be published after the following comments have been addressed.

Page 4 line 148, Please reference the PRISMA guidelines in the text as well as Table 1.

Table 1. The Table would be clearer if the title stated that the ‘reported on page #’ column referred to the paper itself. If found this slightly confusing as I am not familiar with the PRISMA approach. The footnote says it has been reproduced from another source so one expects the whole table to have been reproduced. Also, by introducing context to what had been presented before (e.g. Title and Structured Summary) I wondered what I had inadvertently missed in my reading to that point.

The paper is rich in acronyms, almost all of which I was unfamiliar and thus almost immediately forgot once defined. I would suggest replacing most, if not all, with full names. I found this a problem with Table 4 as many were not re-defined since their first use much earlier in the text. May I issue a friendly challenge to the authors to recall now what OEHAA, HWISE, HWI, SES, SDWA, AHS, WPI, SDWIS all mean?

The discussion from page 20 to 21 could be shortened as there is much repetition of the results when describing each study’s relevance to the situation in Oregon. This would be better if the authors could discuss this within a legislative or policy framework, rather than each study described again and discussed. Indeed, this was done in the recommendations section and the two could easily be combined for brevity. However, within this section the Water Poverty Index was not discussed, and the reader must assume that the authors do not think it suitable for Oregon. The reasons why would be enlightening.

Author Response

Dear Mr. Tian and Reviewer 1,

Thank you for giving us the opportunity to submit a revised draft of our manuscript titled, A Systematic Literature Review on Water Insecurity from an Oregon Public Health Perspective, to the International Journal of Environmental Research and Public Health. We appreciate the time and effort that you and the reviewers have dedicated to providing feedback on the manuscript. We have been able to incorporate changes to reflect most of the suggestions provided by the reviewers. We have highlighted the changes within the manuscript.

Here is a point-by-point response to the reviewers’ comments and concerns.

Comments from Reviewer 1

Comment 1: Page 4 line 148, Please reference the PRISMA guidelines in the text as well as Table 1.

Response: Thank you for mentioning this. We have added an in-text reference to the PRISMA guidelines on Page 4, line 148. It has also been cited at the end of the Table 1 footnote on Page 6, line 155.

Comment 2: Table 1. The Table would be clearer if the title stated that the ‘reported on page #’ column referred to the paper itself. If found this slightly confusing as I am not familiar with the PRISMA approach. The footnote says it has been reproduced from another source so one expects the whole table to have been reproduced. Also, by introducing context to what had been presented before (e.g. Title and Structured Summary) I wondered what I had inadvertently missed in my reading to that point.

Response: We agree that Table 1 might cause confusion for readers who are unfamiliar with the PRISMA approach. Rather than modify the title as suggested, we opt to remove Table 1 from the manuscript (Page 4 and 5, lines 154-157).

Comment 3: The paper is rich in acronyms, almost all of which I was unfamiliar and thus almost immediately forgot once defined. I would suggest replacing most, if not all, with full names. I found this a problem with Table 4 as many were not re-defined since their first use much earlier in the text. May I issue a friendly challenge to the authors to recall now what OEHAA, HWISE, HWI, SES, SDWA, AHS, WPI, SDWIS all mean?

Response: We agree and have incorporated the full name of each acronym at first use, at the beginning of each section throughout the paper. We have also added the full names for the acronyms in Table 4 (now Table 3) after first use within the table.

Comment 4: The discussion from page 20 to 21 could be shortened as there is much repetition of the results when describing each study’s relevance to the situation in Oregon. This would be better if the authors could discuss this within a legislative or policy framework, rather than each study described again and discussed. Indeed, this was done in the recommendations section and the two could easily be combined for brevity. However, within this section the Water Poverty Index was not discussed, and the reader must assume that the authors do not think it suitable for Oregon. The reasons why would be enlightening.

Response: Thank you for your suggestion. We have compared pages 20 and 21 with previous sections to eliminate repetition and identified page 21, second sentence, line 452-453 as a duplicate. We have eliminated the following sentence: The study found that 1.8 to 2.9 billion people worldwide experience severe water scarcity for at least 4 to 6 months per year.  

We agree that it would be fruitful to discuss the Water Poverty Index (WPI) in the Recommendations section; therefore, we have moved content from page 21 (paragraph 2, lines 470-472, 475-477, 482-485) to form a new paragraph discussing the utility of the WPI in Oregon (found in the Recommendations section on page 23, lines 567-574). 

However, since page 20 and 21 adheres to the Summary of Evidence PRISMA checklist item (e.g. summarize the main findings including the strength of evidence for each main outcome; consider their relevance to key groups), we prefer to keep within the structure of these guidelines and maintain a separate Discussion and Recommendation section.

We look forward to hearing from you regarding our submission.

Reviewer 2 Report

This manuscript is a very significant contribution to scholarship on water insecurity and public health, well researched, well organized and well written.  It will be tremendously helpful to all of us who work in this field and I anticipate it will be widely read and cited.  IJERPH should absolutely publish it.

The only suggestion I have is that I would love to see the authors add a specific section summarizing their recommendations for future research, scholarship and public action on water insecurity, beyond the state of Oregon, if they are so inclined.

Author Response

Dear Mr. Tian and Reviewer 2,

Thank you for giving us the opportunity to submit a revised draft of our manuscript titled, A Systematic Literature Review on Water Insecurity from an Oregon Public Health Perspective, to the International Journal of Environmental Research and Public Health. We appreciate the time and effort that you and the reviewers have dedicated to providing feedback on the manuscript. We have been able to incorporate changes to reflect most of the suggestions provided by the reviewers. We have highlighted the changes within the manuscript.

Here is a point-by-point response to the reviewers’ comments and concerns.

Comments from Reviewer 2

 Comment 1: The only suggestion I have is that I would love to see the authors add a specific section summarizing their recommendations for future research, scholarship and public action on water insecurity, beyond the state of Oregon, if they are so inclined.

Response: This is a great suggestion. Thank you. While we would like to offer recommendations for future research, scholarship and public action on water insecurity outside the state of Oregon, at present, the request goes beyond what we perceive to be the scope of the current work. We would like to implement interventions to mitigate water insecurity risk in Oregon before we make recommendations elsewhere.

We look forward to hearing from you regarding our submission.